# Production of *Sitobion avenae*-resistant *Triticum aestivum* cvs using laccase as RNAi target and its systemic movement in wheat post dsRNA spray

**Asma Rafique**[1], **Amber Afroz**[1]*, **Nadia Zeeshan**[1], **Umer Rashid**[1], **Muhammad Azmat Ullah Khan**[1], **Muhammad Irfan**[1], **Waheed Chatha**[1], **Muhammad Ramzan Khan**[2], **Nazia Rehman**[2]

1 Department of Biochemistry and Biotechnology, University of Gujrat, Gujrat, Punjab, Pakistan, 2 National Institute for Genomics and Advanced Biotechnology, National Agricultural Research Centre, Islamabad, Pakistan

* dramber.afroz@uog.edu.pk, ambernics01@gmail.com

**Data Availability Statement:** All relevant data are within the paper and its Supporting Information files.

## Abstract

Among the wheat biotic stresses, *Sitobion avenae* is one of the main factors devastating the wheat yield per hectare. The study's objective was to find out the laccase (lac) efficacy; as a potential RNAi target against grain aphids. The *Sitobion avenae* lac (Salac) was confirmed by Reverse Transcriptase-PCR. Gene was sequenced and accession number "ON703252" was allotted by GenBank. ERNAi tool was used to design 143 siRNA and one dsRNA target. 69% mortality and 61% reduction in lac expression were observed 8D-post lac DsRNA feeding. Phylogenetic analysis displayed the homology of grain aphid lac gene with peach potato, pea, and Russian wheat aphids. While Salac protein was found similar to the Russian grain, soybean, pea, and cedar bark aphid lac protein multi-copper oxidase. The dsRNAlac spray-induced silencing shows systematic translocation from leaf to root; with maximum lac expression found in the root, followed by stem and leaf 9-13D post-spray; comparison to control. RNAi-GG provides the Golden Gate cloning strategy with a single restriction ligation reaction used to achieve lac silencing. *Agrobacterium tumefaciens* mediated *in planta* and *in-vitro* transformation was used in the study. *In vitro* transformation, Galaxy 2012 yielded a maximum transformation efficiency (1.5%), followed by Anaj 2017 (0.8%), and Punjab (0.2%). *In planta* transformation provides better transformation efficiencies with a maximum in Galaxy 2012 (16%), and a minimum for Punjab (5%). Maximum transformation efficiency was achieved for all cultivars with 250 µM acetosyringone and 3h co-cultivation. Galaxy 2012 exhibited maximum transformation efficiency, and aphid mortality post-feeding transgenic wheat.

## Introduction

*Triticum aestivum* L. is the second most important protein source of human nutrition; staple food consumed by more than 35% world's population [1]. Wheat production is lower than

**Funding:** A Afroz NRPU6506 National Research Program for Universities (NRPU), Higher Education Commission (HEC), Islamabad Pakistan. https://www.hec.gov.pk/english/services/universities/nrpu/Pages/Introduction.aspx. No, The funders had no role in study design, data collection and analysis, decision to publish, or preparation of the manuscript.

**Competing interests:** "The authors have declared that no competing interests exist"

**Abbreviations:** RNAi, RNA interference; dsRNA, double-stranded RNA; siRNA, small interfering RNA; RISC, RNA-induced silencing complex; RT-PCR, Reverse transcriptase polymerase chain reaction; qRT-PCR, Quantitative reverse transcriptase polymerase chain reaction; MSA, Multiple sequence alignment; Salac, laccase (lac), *Sitobion avenae* Laccase.

demand, attributed to improper agriculture practices, and abiotic, and biotic stresses including weed, and insect infestations [2]. Aphids are responsible for significant economic losses through feeding directly, and indirectly as virus carriers [3]. *Sitobion avenae* is the most reported grain aphid throughout the world; for severe yield losses [4, 5]. Chemical treatments are deleterious, and transgenic approaches are limited due to insecticidal toxin tolerance in plants. RNAi was an alternative to silencing the basic gene transcripts. The route is substantially conserved in aphids, resulting in successful salivary gene knockdown in pea, peach, and English grain aphids [6, 7]. DsRNA or siRNA-mediated gene silencing had been successfully reported in many insect species which is variable. The reason is dsRNA degradation/uptake, RNAi selected genes efficacy, non-novel RNAi target, and viral infection interference [8]. Short interfering RNA (siRNA) targeting COO2: the most abundant salivary gland transcript in pea aphids, caused silencing and elevated insect mortality are used as <u>+ve</u> control [9].

In aphid species, different genes expressed in the gut, salivary gland, and embryos had been identified as effective RNAi targets causing death, growth reduction, and aphid sterility [5]. DsRNA generated *in-vitro*, siRNA, artificial feeding, and microinjection had been used for gene knockdown. For systemic RNAi salivary gland targets with dsRNA feeding are specifically used [5]. Laccase (lac) protein is found in the aphid's salivary glands that regulate the glycosyl phosphatidyl inositol-anchor biosynthesis pathway along with lipid biosynthesis [10]. It had been identified in plants' defense breakdown by lignocellulose digestion by insects with the disturbance of their metabolic homeostasis or linked to insect defense by its cuticle hardening for resistance to insecticides [10–14]. The cell wall strength is a part of sequential processes for plant immunity development including the pathogen-associated molecular pattern-induced immunity after the insect aphid attack induced by saliva [15].

*A. tumefaciens*, naturally occurring, gram-negative soil bacterium-mediated transformation is preferred in plants; as it results in higher transformation efficiency with low cost rather than any other method [16]. The transformation is dependent on efficient T-DNA delivery; mediated by acetosyringone [17]. *Agrobacterium*-mediated transformation system was optimized in wheat and rice mature seeds with optimized acetosyringone concentrations [18, 19]. The lines produced from durum, bread wheat, barley, and maize express a single copy number [20]. Co-cultivation period optimization is linked to higher transformation efficiency [19]. *In-vitro Agrobacterium*-mediated transformation efficiency was found in a range from 1.28–1.77% [21].

The Golden Gate (GG) cloning strategy was used in this work to adapt the previously reported pRNAi-GG vector for higher silencing [22]. Previous methods for creating hairpin RNA were multi-step, expensive, and time-consuming procedures [23]. The present study discusses the simple methodology for ihpRNA construct for *S. avenae* laccase (Salac1) gene silencing resulting in transgenic wheat. The amplified PCR products for Salac1 silencing were produced in a single-step restriction-ligation reaction in the RNAi-GG vector. The lac was sequenced in native *S. avenae*, and its potential as a dsRNA target by artificial feeding, spray application, and transgenic approaches was tested. Systemic movement of the dsRNA in the root, shoot, and leaves of wheat 9-13D post spray was tested as well. ERNAi tool was used to find siRNA and dsRNA targets from the lac sequence with no off-targets. The vector was transformed in *Triticum* cvs using *Agrobacterium*-mediated *in-planta* and *in-vitro* transformation; followed by insect bioassay.

## Materials and methods

### Research approval

The Research is approved by the Department of Biochemistry and Biotechnology, Research and Review Committee (DRRC), and Directorate of advanced studies and research board

(ASRB), University of Gujrat. Research work is performed in the Department of Biochemistry and Biotechnology, University of Gujrat, Gujrat Pakistan.

## Aphid sample collection and rearing

*S. avenae* was collected from wheat fields in Gujrat and reared on *Triticum aestivum* cv. Faisalabad 2008 under standard conditions (White fluorescent light 300 μmol/m$^2$/s, 16h light/8h dark at 22˚C, and 70% relative humidity). Aphids were kept continuously parthenogenetically reproducing clones. 3-5D-old adult aphids were collected for RNA extraction.

## RNA extraction and cDNA synthesis

Total RNA was isolated from *S. avenae* using an RNA extraction Kit according to the manufacturer's protocol (12183018A Pure Link™ RNA Mini Kit). The cDNA was synthesized from RNA with Revert Aid First Strand cDNA Synthesis Kit (Cat #: K2563; Thermo Scientific™).

## Reverse transcriptase polymerase chain reaction (RT-PCR)

Lac primers were designed (using Primer-BLAST) and validated using an oligonucleotide property calculator; which shows no self-complementarity or hairpin formation. The PCR reaction contains 1.5μL cDNA, 1 μL MgCl$_2$, 0.5 μL dNTPs mix, 1μL forward, and reverse primer (10 μM final concentration), 1μL Taq buffer, 0.5μL Taq polymerase, with a final volume of 25 μL. The PCR program involved 5 min initial denaturation at 95˚C, followed by 40 cycles of denaturation (95˚C for the 30s), annealing (52˚C for the 30s), extension (72˚C for 60s), and a final 5min extension at 72˚C. The PCR product was separated on 1% Agarose gel.

## Gene cloning

The target sequence was amplified using primers that includes the T7 promoter sequence at the 5′ ends of both forward, and reverse primers. The amplified PCR fragment was eluted, and purified with a Gel Extraction kit, and cloned into the pTZ57R/T plasmid (Fermentas InsTA clone PCR Cloning Kit #K1214) according to the manufacturer's protocol. The heat shock method was used to transform the plasmid into *E. coli* (DH5α). LB agar plates selection was with ampicillin and LacZ. After 24h, white recombinant colonies were screened. The plasmid was extracted (GeneJET plasmid Miniprep kit #K0502) from recombinant colonies, and confirmed by colony PCR.

## Sequencing, and phylogenetic analysis

After purification, the plasmid was sent to Macrogen Korea for sequencing. BLAST was used to analyze sequencing data which was then followed by phylogenetic analysis (phylogene.fr). ERNAi tool was used to design potent siRNA targets against the lac1 mRNA sequence with *A. pisum* as standard (Boutros lab, E-RNAi-Version 3.2) [23].

## dsRNA assay

For the *S. avenae* artificial diet, 20% sucrose (pH 7.2), with 20 ngμL$^{-1}$ dsRNA was used [22]. dsRNA of lac and GFP was produced with T7 promoter sequence (`TAATACGACTCACTA TAGGG`) at the 5' ends of both primers with MEGAscript RNAi kit (Ambion Huntingdon, UK# AM1626). The number of aphids fed on dsRNA was counted from 1-8D.

**Table 1. List of primers for PCR, qRT-PCR. and dsRNA assays.**

| Primer | Sequence | Tm |
|---|---|---|
| Lac1 FP | 5′-ATGTCATATGATTTTTATTC-3′ | 53˚C |
| Lac1 RP | 5′-CCTCAACGTGGAACTCAA-3′ | |
| Lac1+T7 FP | 5′-TAATACGACTCACTATAGGGATGTCATATGATTTTTATTC-3′ | 53˚C |
| Lac1+T7 RP | 5′-TAATACGACTCACTATAGGGCCTCAACGTGGAACTCAA-3′ | |
| Actin FP | 5′-GGTGTCTCACACACAGTGCC-3′ | 60˚C |
| Actin RP | 5′-CGGCGGTGGTGGTGAAGCTG-3′ | |
| GFP+T7 FP | 5′-GGATCCTAATACGACTCACTATAGGAAGAGTGCCATGCCCGAAGGT-3′ | 60˚C |
| GFP+T7 RP | 5′-GGATCCTAATACGACTCACTATAGGAAAGGACAGGGCCATCGCCAA-3′ | |
| RNAi-GG FP | 5′-AACAAGATGGATTGCACGCA-3′ | 58˚C |
| RNAi-GG RP | 5′-GCAATATCACGGGTAGCCAA-3′ | |
| Lac1-FP+Adaptor | 5′-ACCAGGTCTCAGGAGCAGAATTAGAAGACGCAACA-3′ | 50˚C |
| Lac1-RP+Adaptor | 5′-ACCAGGTCTCATCGTCCTCAACGTGGAACTCAA-3′ | |
| P-21 | 5′-ACCATTTACGAACGATAGCC-3′ | 55˚C |
| P-22 | 5′-GTAAAACGACGGCCAGTG-3′ | 55˚C |
| P-24 | 5′-CATTTTAGCTTCCTTAGCTCC-3′ | 55˚C |
| P-25 | 5′-CATTTGGATTGATTACAGTTGG-3′ | 56˚C |

## Quantitative reverse transcriptase polymerase chain reaction (qRT-PCR)

qRT-PCR was used to analyze the expression level of the Salac1, and actin gene 1-8D-post feeding (Table 1). The qRT-PCR reaction was carried out using the Step One™ Real-Time PCR System using the SYBR Green PCR Core Reagent (Cat. No. 4309155; Thermo Fischer Scientific) as directed by the manufacturer. The total reaction mixture contains 1μL cDNA, 0.5μL primer (10 pmol/μl), and 10μl 2x SYBR Green Supermix reagent. For all PCR reactions, the PCR program used was: 95˚C for 4min, then 40 cycles of 95˚C for the 30s, 60˚C for 30s, 72˚C for 30s, and 10 min at 72˚C at the end. The lac primer efficiency is calculated by sample serial dilutions. Ct values are obtained and plotted against the logarithm of substrate concentrations. A linear regression curve was drawn to calculate the slope of the trend line, and primer efficiency is calculated by: $E = (10^{(-1/slope)}-1)100$ (S2 File). The amplification efficiency found was 109.5%. Ct value was compared to dsRNA treated samples, transgenic wheat plants, along with control for the sample quantitation.

## dsRNA spray application

The second leaves of 3-W-Old *T. aestivum* cv Faisalabad 2008 were detached and placed on Petri plates with 1% agar. The dsRNA (500 ng μL$^{-1}$) was mixed with TE Buffer (40uM Tris + EDTA) for a final concentration of 20 ng μL$^{-1}$ [22]. TE buffer (same concentration) was **-ve** control; while **+ve** control was 20 ng μL$^{-1}$ actin-dsRNA. Leaf spray was done on the leaf's upper surface. Each plate is with 10 detached leaves; the lower leaf surface was covered with plastic. Treated leaf samples are placed in a growth chamber. 2D-post-spray; aphids were placed on the non-sprayed part of each leaf using clip cages. To measure the amount of sprayed dsRNA in different tissues (first leaf, second leaf, shoot, and root), samples were taken 9, 11, and 13D-post spray treatment of the first leaf.

## Vector construct

pRNAi-GG was used to generate the ihpRNA construct for Salac1 silencing. RNAi-GG vector is a binary vector (15,796 bp): having Neomycin and Kanamycin resistance gene for bacterial

selection, two copies of ccdB flanked with Bsa1, and Pdk intron [24] (Fig 3). In the vector, the Salac1 was placed under the control of the CAMV 35S promoter at the Bsa1 restriction site by replacing ccdB. The binary vector was constructed having the right and left borders, the origin of replication, selectable markers, and Salac1 (S1 File).

### Insert confirmation

After RNAi-GG plasmid isolation, and confirmation with colony PCR, *S. avenae* RNA was extracted using the PureLink® RNA Mini Kit technique according to the manufacturer protocol followed by cDNA synthesis using Thermo Fisher Scientific's Revert Aid first strand cDNA synthesis kit (#k1622). Primers; were designed; in addition to the sequences of lac1, the restriction site of enzyme BsaII and the 5" end an adapter sequence was added; for laccase insertion in the vector (Table 1). For PCR amplification, gene-specific primers of Salac1 were used. GG cloning system uses a single restriction ligation reaction which includes the 5 μL RNAi-GG plasmid, 2 μL purified PCR product (lac1 with RNAi-GG specific adaptor sequences), 1 μL 10x ligation buffer, 0.5 μL T4 DNA ligase (Solarbio Kit Cat. 10481220001), 0.5 μL BsaI (Solarbio Kit Cat. Q2020), and 1μL ligation buffer.

### *E. coli*, and *Agrobacterium tumefaciens* transformation

DH5α single colony was transferred to 50 mL LB broth, and incubated (37˚C), at 250 rpm (OD = 0.4–0.6). DH5α (200 μL) competent cells and 5μL recombinant RNAi-GG plasmid were transformed by the heat shock method [25]. The selection was through 50 mg $L^{-1}$ kanamycin. The transformed colonies were confirmed by restriction analysis, and colony PCR. *A. tumefaciens* (LBA4404) transformation involved plasmid isolation from DH5α positive colonies (Thermo Scientific GeneJET Plasmid Miniprep Kit); confirmed by restriction analyses, and PCR. LBA4404 cells were grown for 36h at 28˚C in an LB selection medium with hygromycin (25 mg $L^{-1}$). A single colony of LBA4404+ recombinant RNAi-GG plasmid (10 μL) was used for transformation. Colony PCR was used for RNAi-GG transformation confirmation. A single colony was lysed at 95˚C mixed with 1 μL $MgCl_2$, 0.5 μL dNTPs mix, 1 μL Salac1 forward, and reverse primer (10 μM), 1μL Taq buffer, 0.5μL Taq polymerase for a final volume of 25 μL. The PCR consisted of initial denaturation (95˚C: 5 min), followed by 40 cycles of denaturation (95˚C: 45s), annealing (52˚C: 45s), extension (72˚C: 1 min), with a final 10min extension (72˚C). The PCR product was run on 1% Agarose gel.

### *In planta*, and *in vitro* transformation

**Callus induction, and infection.** 3D-Old *T. aestivum* cvs Anaj 2017, Galaxy 2012, and Punjab seeds were surface sterilized with Clorox (sodium hypochlorite 60%) for 15min. MS salt+ Vitamins+ (3 mgL$^{-1}$ of 2, 4-Dichlorophenoxyacetic acid, 20% coconut water) was used for callus induction [25]. The 3-W-Old callus is used for *Agrobacterium* infection and transformation. An inoculation medium containing bacterial culture was applied to the embryogenic calli and incubated for 3h at 28˚C.

**Inoculum preparation.** The 40 mL LB broth was added to the actively growing *A. tumefaciens* (LBA4404) with recombinant RNAi-GG vector (200μL): and placed at 28˚C for 48h until optical density (O.D.)$_{600nm}$ reached 0.4–06. The culture was centrifuged at 4000 rpm for 15min. The pellet was resuspended in the Amino Acid medium [26], with acetosyringone (50, 100, 150, 200, 250, and 300 μM). This suspension was used for seedlings, and callus infection. The infected calli were placed on Petri plates moistened with MS Medium+ Vitamins+ (50, 100, 150, 200, 250, and 300μM) acetosyringone filter papers and incubated for 3h (28˚C) in the dark for co-cultivation.

**Selection regeneration.** Following co-culture for 3h, calli were washed with cefotaxime (500mgL$^{-1}$) containing callus induction medium [19]. The calli after drying were transferred to the selection regeneration medium (MS salts+ vitamins+ 1.5 mgL$^{-1}$ BAP, 10% coconut water, 500 mgL$^{-1}$ cefotaxime+ 50 mg L$^{-1}$ Hygromycin) [25]. Regenerating calli were shifted to fresh medium after 2W. The transgenic shoot was shifted to the rooting selection medium (MS salts+ vitamins+ 0.5 mgL$^{-1}$ IAA, 500 mgL$^{-1}$ cefotaxime+ Hygromycin 50 mg L$^{-1}$). The T$_o$ plants were raised to produce T$_1$ plants; tested for lac confirmation.

**DNA extraction, PCR, and qRT-PCR analysis.** The genomic DNA was extracted from T$_1$ generation produced through *in planta* transformation [27]. The PCR reaction was performed using the same conditions as described above. The PCR product was run on 1% Agarose gel. qRT-PCR was used to analyze the transformed Salac1 expression level by conditions as discussed above.

**Insect bioassay.** For the *S. avenae* bioassay: aphids were reared on resistant, and susceptible wheat cultivars. The transgenic, and non-transgenic (control) *Triticum* cvs; Punjab, Galaxy 2012, and Anaj 2017 were used; with Zincol 2016 (Resistant cv) as **+ve** control. To assess Salac1 silencing by *In planta* transformation, the reproduction, and survival rate of *S. avenae* in all cvs with control were monitored.

## Statistical analysis

A set of 3 experiments are used to determine the standard error in MS Excel. Minitab was used to determine the analysis of variance (ANOVA) of aphid's mortality and mRNA expression 2, 4, 6, and 8D-post dsRNA feeding (20 ngL$^{-1}$), **-ve** control (20% Sucrose), and internal control. Different alphabets, and * show significant, while ** show highly significant differences from the control.

## Results

### Screening for insects and PCR amplification

*S. avenae* was identified as apterous adult aphids (2mm long) with a green/brown body; black legs, black antennae emerging from frontal lobes, and swelling in the middle (Fig 1A). RNA-produced cDNA was used for lac PCR analysis producing a 615 bp band (Fig 1B). The amplified gene was purified, inserted into the pTZ57R/T vector, and blue-white screening was used for transformants by colony PCR. Plasmid DNA was prepared was sent for sequencing. The lac sequence was submitted to GenBank and given the ID ON703252.

### Phylogenetic analysis, & siRNA target prediction

The phylogenetic tree shows 90% homology with *M. persicae*, *A. pisum*, and *Diuraphis noxia*, (90%) with 95% coverage (Fig 1C). The lac protein domain was similar to the *Aphis craccivora*, *Tribolium castaneum*, *Dinoponera quadriceps*, *Neodiprion lecon*, *A. pisum*, and *A. glycines*, copper oxidase copper-binding domain. ERNAi tool was used to design potent siRNA targets against the lac1 mRNA sequence in *A. pisum*. 143 siRNA and one dsRNA target are designed; can be used against grain aphids as well as the related species identified (S1 Table).

### dsRNA artificial diet evaluation and qRT-PCR analysis

*S. avenae* artificial diet containing 20% sucrose, was used as an artificial diet, with 20 ng μL$^{-1}$ dsRNA [22]. 8D-post-feeding, 69% mortality was found compared to **+ve,** and **-ve** control (Number of replicates/treatment was n = 25, p<0.05) (Fig 2A). qRT-PCR revealed a 61% lac expression reduction; 8D-post-feeding compared to control (dsGFP) (Fig 2B).

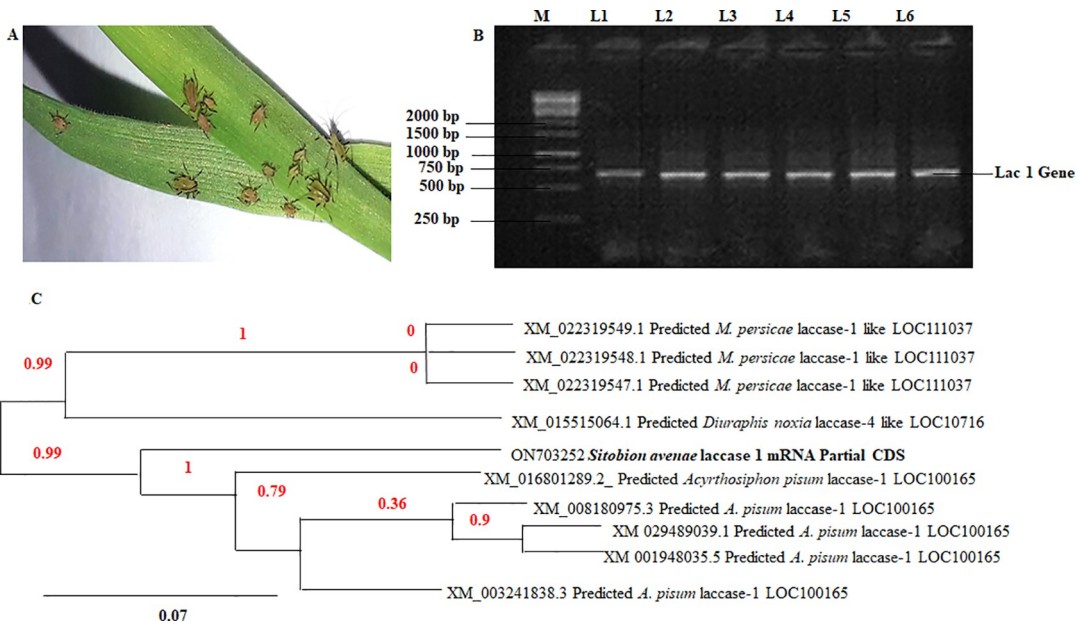

**Fig 1.** (**1A**) Sampling and screening of apterous *Sitobion avenae* on 3-week-old wheat plants. (**1B**) Reverse Transcriptase Polymerase chain reaction for (2A) Lac gene (615 bp) (L1) 1 Kb ladder, (L2) Lac gene. (**1C**) Phylogenetic tree of Lac gene (ON703252: *S. avenae* lac1 mRNA Partial CDS) closest homology (≥90%) by phylogeny.fr show similarity to *M. persicae*, *A. pisum*, and *Diuraphis noxia*.

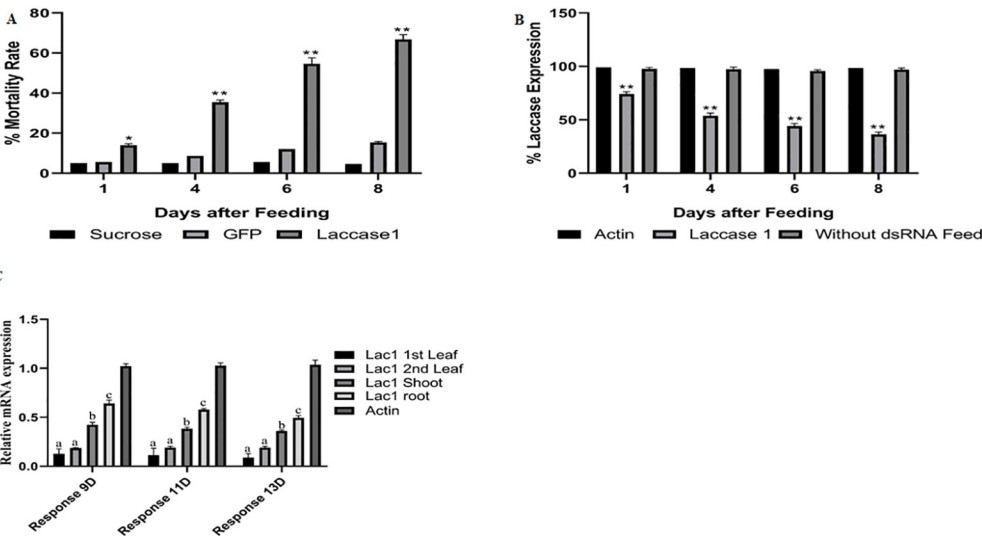

**Fig 2.** (**2A**) DsRNA feeding experiment for checking the efficacy of lac gene as RNAi target. Effect of artificial diet (20% sucrose), GFP dsRNA (20 ngL$^{-1}$) **-ve** control, and Lac gene (LAC) (7 μg/μL) (**2B**) Fold increase in mRNA expression (determined by of Quantitative reverse transcriptase polymerase chain reaction (qRT-PCR) of Lac gene in comparison to internal control (Actin) and -ve control (Lac without dsRNA feeding assay) 2D, 4D and 6D post-feeding. (**2C**) Fold change in mRNA expression in leaf, stem and root; determined by Lac gene qRT-PCR in comparison to internal control (Actin) 9D, 11D and 13D post-spraying. Means are significantly different at a 0.05 level of significance at α = 0.05.

### dsRNA mediated gene silencing in *S. avenae* reared on wheat leaves

Systematic RNA silencing had been reported to move across the vascular systems [28, 29]. Aphids feeding on transgenic potatoes express dsRNA and produce significantly lower mRNA levels in comparison to the control [22]. Locally sprayed lac dsRNA confers gene silencing in grain aphids feeding from, non-sprayed segments of the same wheat leaf. After 9, 11, and 13D-post dsRNA spray; Galaxy 2012 1st, 2nd leaf, shoot, and root RNA was checked for dslac translocation using qRT-PCR. The relative expression level of the aphid's lac gene was maximum in the root (60%), followed by the shoot (40%) reduced to (20%) in 2nd leaf; compared to the aphids feeding on the–**ve**, and **+ve** control (TE Buffer) 9-13D-post feeding (Fig 2C).

### RNAi-GG vector with lac gene confirmation

A colony PCR with RNAi-GG-specific primers from the pdk intron region shows a 680 bp band at Tm = 58 (Table 1; Figs 3 and 4A). The lac1 gene was confirmed with lac1 with vector-specific primers including adaptor sequences at Tm = 53, and 645 bps is obtained (Fig 4B). A single restriction ligation reaction was performed for lac1 silencing, transformation in sense, and antisense orientation flanked on both sides of the pdk intron. Purified lac1 PCR product, RNAi-GG vector, BsaI, and T4 DNA ligase were incubated for restriction, followed by the transformation in DH5α. RNAi-GG restriction analysis with SacI, and SwaI show 2948, and 12824 bp fragments confirmed lac1 integration (Figs 3 and 4C). P21 and P22 primers confirmed the whole insert (lac1 sense+ antisense strand on both sides of pdk) with a 3194 bp band at Tm = 55˚C (Fig 4D). The lac1 sense strand was confirmed with 1000bp band with P-21, & P-24 primers at Tm = 55˚C (Fig 5A). The lac1 antisense strand was confirmed with P-22 + P-25 primers by a 958 bp band at Tm = 56˚C (Fig 5B). The lac1 antisense strand orientation was confirmed with P-22+ lac1 reverse primer for the 913 bps band at 56˚C (5C).

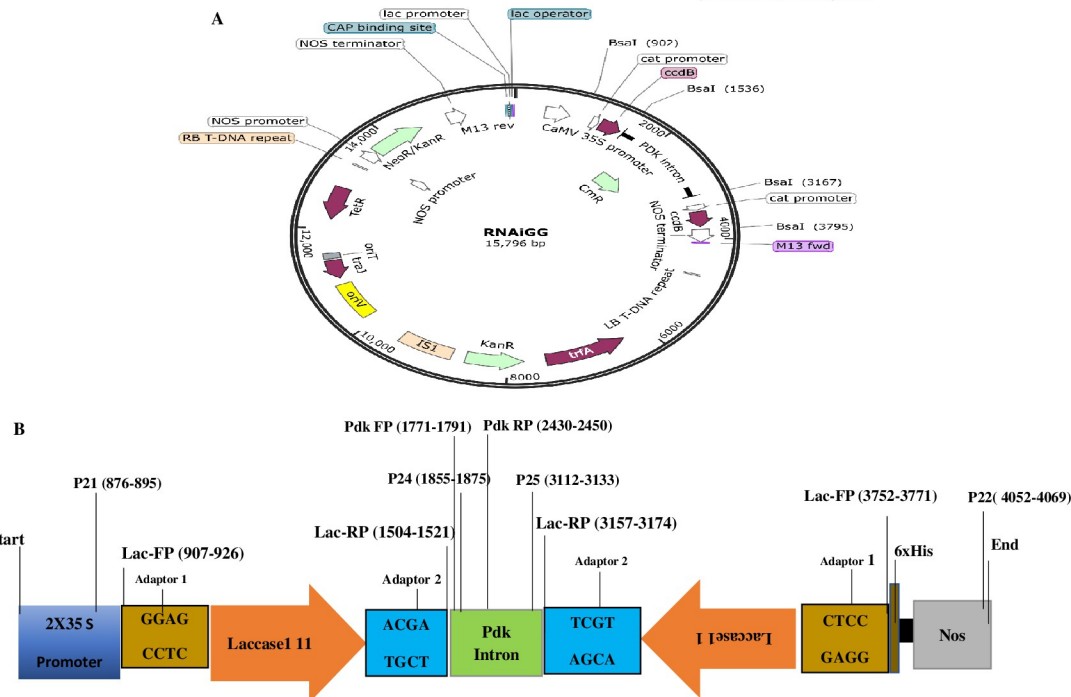

**Fig 3. Schematic diagram of the SaLac1 location in the ihpRNA vector.** The CaMV 35S promoter, two copies of the SaLac1 gene, the Pdk intron, Bsa1, and an adapter sequence, and NOS terminator form the cassette.

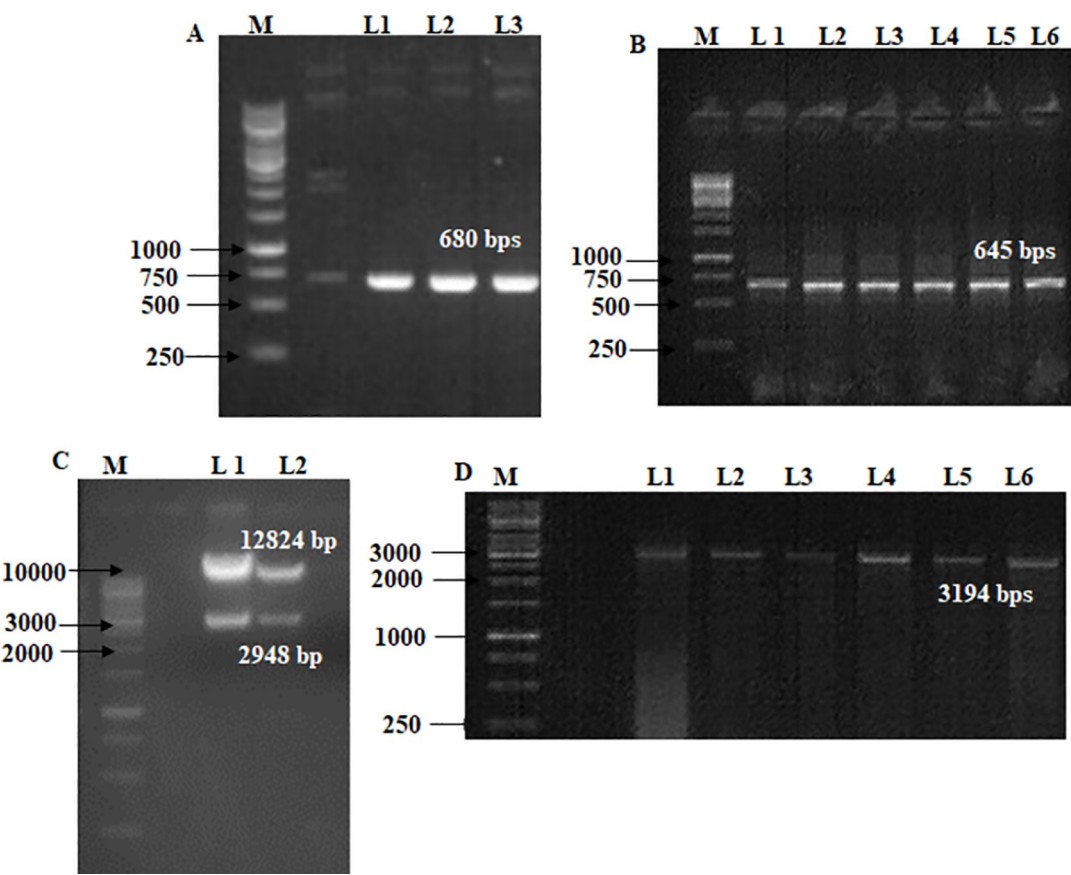

**Fig 4.** (**4A**) Colony PCR of RNAi-GG with RNAi-GG specific primers from pdk intron region (Tm. 58; 680 bps). M (1kb Ladder), L1-3 (PCR products of RNAi-GG). (**4B**) Lac1 confirmation with vector-specific primers including adaptor sequences. M (1kb Ladder), L2-6 (PCR Product: 645 bps at Tm = 53) (**4C**) RNAi-GG digestion with SacI, and SwaI (12824, and 2948 bps): M (1kb Ladder), L1-2 (Digestion products) (**4D**) PCR product using P-21 + P22 primers for whole insert confirmation (3194 bps), M (1kb Ladder), L1-6 (Whole insert: 3194 bps; Tm = 55).

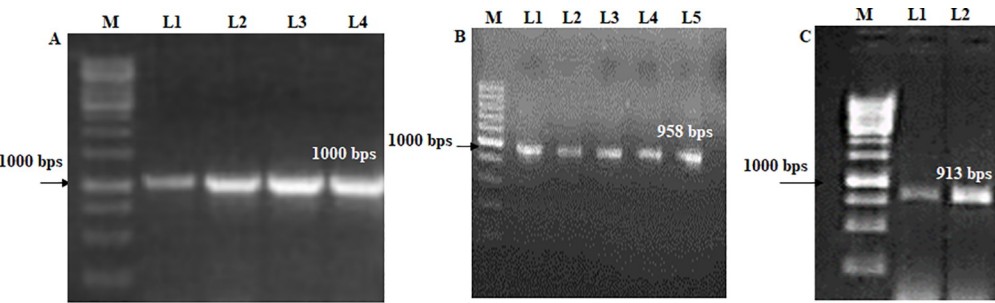

**Fig 5.** (**5A**) Lac 1 sense orientation confirmation by P-21 & 24 primers. M (1kb Ladder), L1-4 (Lac1 sense insert) (Tm = 55; 1000 bps) (**5B**) Lac 1 antisense orientation confirmation by P-22 and P-25 primers. M (1kb Ladder), L1-5: Lac1 antisense insert (Tm = 56; 958 bps) (**5C**) Lac1 antisense orientation confirmation by P-22 and Lac1 Reverse Primer. M (1kb Ladder), L1-2 (PCR antisense orientation insert) (Tm = 56; 913 bps).

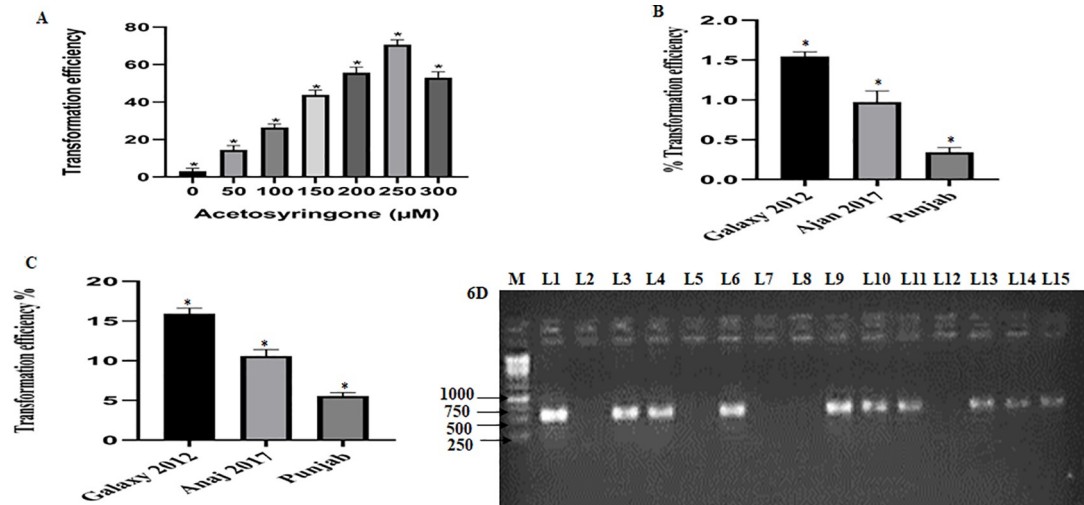

**Fig 6.** (**6A**) Effect of acetosyringone concentration on transformation efficiency (**6B**) *T. aestivum* cvs Galaxy 2012, Anaj 2017, & Punjab transformation efficiency by *In Vitro* Assay (**6C**) *T. aestivum* cvs Galaxy 2012, Anaj 2017, & Punjab transformation efficiency by *In planta* Assay (**6D**) Lac 1 confirmation T1 *Triticum aestivum* cultivars. M (1kb Ladder), L1 (+ve control), L2 (-ve control), L3-4, 6 (Anaj 2017), L9-11 (Galaxy 2012), L13-15 (Punjab), L7, L8, & L12 (non-transgenic plants).

### *Agrobacterium* transformation

After lac1 confirmation in RNAi-GG vector, its sense, the antisense strand, and antisense strand orientation: Rec RNAi-GG was transformed in *Agrobacterium* (LBA4404); confirmed by colony PCR. Seedlings after vernalization for 20D (4°C) were shifted to pots. $T_1$ *Triticum* cvs (Anaj 2017, Galaxy 2012, and Punjab) were confirmed for Salac1 by RT-PCR (Fig 6D). 250 μM acetosyringone was found best for transformation (70%) (Fig 6A). Acetosyringone is a phenolic compound that acts as a stimulator of vir genes [19]. Without the addition of aceto-syringone, no transformation was observed.

### Transformation efficiency

Co-cultivation for 3h with 250μM acetosyringone resulted in maximum transformation efficiency (Fig 6A). The transformation efficiency was calculated 15D post selection with hygro-mycin (50 mgL⁻¹) in Galaxy 2012 callus. The maximum selection was observed with 250 μM acetosyringone. After callus selection, and regeneration, the plants were transplanted to soil; regenerated plants were grown to maturity, and all the $T_o$ plants were morphologically normal. Putative transgenic plants were confirmed by PCR amplification of the Salac1 gene. All geno-types tested showed transformation efficiencies significantly different from one another. The maximum transformation efficiency was of Galaxy 2012 (1.5%), followed by Anaj 2017 (0.8%), and Punjab (0.2%) (Fig 6B).

Tissue culture encounters a series of difficulties in regeneration and contamination of *in-vitro* cultures thus producing less number of transformants [30]. A continuing series of subcul-tures on an antibiotic-containing medium serves as the selection step for tissue culture trans-formation. After callus co-culture with the bacterial strain in tissue culture, cefotaxime was employed to limit the development of *Agrobacterium*. Some researchers have suggested that the poor success rate of wheat transformation may be owing to antibiotics' potential toxicity to callus development [31]. *In-vitro* transformation efficiency was low, so for higher transforma-tion efficiency: *in planta* transformation was employed.

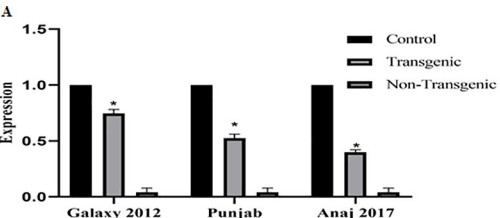
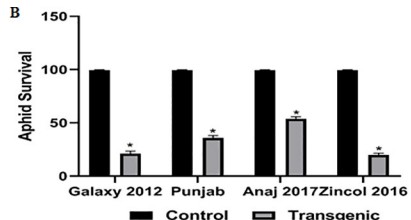
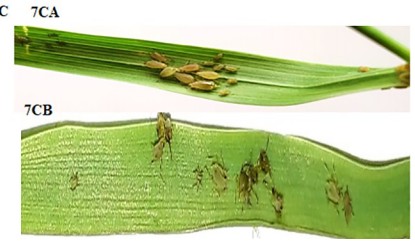

**Fig 7. (7A)** qRT-PCR relative lac 1 expression in *In planta* transformation. Actin was used as control. Data is presented as mean ±SE of the mean. *indicates a significant difference in Laccase1 expression in transgenic Galaxy 2012, Punjab, and Anaj 2017. No Laccase1 expression in non-transgenic Galaxy 2012, Punjab, and Anaj 2017. **(7B)** Insect bioassay on *T. aestivum* resistant cv. Zincol 2016, and Susceptible cvs. (Galaxy 2012, Anaj 2017, & Punjab). **(7C)** Insect bioassay on *T. aestivum* transgenic T1 cv. Galaxy 2012 **(7CA)** Before feeding and **(7CB)** After Feeding.

### *In planta* transformation

*In planta* transformation provides stable, efficient, and quick transformation [30]. Three susceptible *Triticum* cvs. Anaj 2017, Galaxy 2012, and Punjab were transformed with LBA4404 harboring RNAi-GG vector. 50 seeds of each genotype were taken and placed in pots after verbalization, and the $T_1$ generation was raised. Lac integration was confirmed by PCR (615 bps) for *Triticum* cvs, with control (Fig 6D). The transformation efficiency of Salac1 was: Galaxy 2012 (16%); Anaj 2017 (10%), and Punjab (5%) (Fig 6C). Higher transformation efficiencies were observed for *in planta* transformation; compared to *in vitro* transformation with tissue culture.

### Insect bioassay, & qRT-PCR

*T. aestivum* cvs. Anaj 2017, Galaxy 2012, Zincol 2016, Ujala 2016, and Punjab were used, out of which three cvs. Anaj 2017, Galaxy 2012, and Punjab were susceptible to *S. avenae* according to antibiosis results (Data not included). Adult *S. avenae* were kept on transgenic, and non-transgenic wheat cultivars Anaj 2017, Galaxy 2012, Punjab, and **+ve** control Zincol 2016 to check *S. avenae* resistance in transgenic *Triticum* cvs. Transgenic wheat cv. Galaxy 2012 shows the lowest aphid survival (22.6%), followed by Punjab (37.6%), and Anaj 2017 (52%); compared to **+ve** control Zincol 2016 (20%) (Fig 7B and 7C). The aphid survival rate indicates the Salac1 silencing is comparable to the resistant Zincol 2016. qRT-PCR shows the highest laccase expression (70%) in Galaxy 2012, followed by Punjab (50%), and Anaj 2017 (30%), compared to no expression in non-transgenic plants. While actin (internal control) shows 100% expression for all cultivars (7A).

### Discussion

ihpRNA constructs are easier to follow but need to be optimized. Primarily ligase-based vectors are produced (pHANNIBAL, pKANNIBAL); with several rounds of restriction and ligation [32]. The pHELLSGATE, and pIPK RNAi vectors: based on the GATEWAY cloning; had

been used. The technique allows the recombination of primers with attB1 and attB2 sites in vector, resulting in hairpin double arms forming a single PCR [33]. But this requires BP and LR reactions which makes it expensive. It was followed by one step overlap extensions PCR method for ihpRNA construct by inverted repeats gathering with an intron in a single tube. The last development was ligation-independent cloning; with one-step cloning, and 2 rounds of PCR [34]. GG cloning was a more straightforward system; that utilizes type IIs restriction enzymes, with the ligation of two digested segments [23]. It offers a simple restriction-ligation process; with a single PCR product of the target gene transformed into a plant ihpRNA expression vector for siRNA-mediated silencing [35]. GG cloning-based assays included shuffling of the target gene/enzyme from different sources for recombination's resulting in a set of efficacies [36], a set of DNAs for recombinant DNA molecules important in biotechnology [37], gene editing by transcription activator-like effector nuclease [38], rice resistance to sugarcane mosaic virus [39]. Especial care is needed so that IIs nuclease site should not be present within the target gene. The recombinants generated b/w PCR products and the RNAi GG; sub-cloning of PCR product is unnecessary and further enhances the high throughput ihpRNA construct.

Pdk intron primers show 680 bps; and lac primers with adapter sequences display a 645 bps band (Figs 3, 4A and 4B). A single restriction ligation reaction was performed for lac1 silencing. Type II restriction endonuclease BsaI is used for restriction and T4 DNA ligase for ligation. P21 and P22 primers confirmed insert (lac1 sense+ antisense strand on both sides of pdk) with 3194 bps (Fig 4D). Sense and the antisense strand were confirmed with 1000 bps, and 958 bp bands; while antisense orientation was confirmed with the 913 bps band (Fig 5A–5C). ihpRNA separated by pdk intron region with type IIs restriction enzyme BsaI in one restriction ligation step is reported [21, 29]. The RNAi had been used to interfere sheath of *S. avenae* [40], ecdysone receptors genes EcR, USP *S. avenae* in wheat [41], and insects [42]. The spacer sequence between the inverted repeats used for GG cloning is necessary for the stabilization of recombinant plasmids [43]. Synthetic intron sequence in *Arabidopsis* is adaptable for the ihpRNA vector resulting in an effective decrease target gene's expression for the mutant phenotype [23]. The wheat mosaic virus NIa protease gene was targeted by ihpRNA having pdk intron between the sense, and antisense strands [44].

The Salac1 had been successfully used for the resistance against grain aphids in wheat using dsRNA assay, and *Agrobacterium*-mediated ihpRNA based *in vitro* and *in planta* transformation. DsRNA as a bio-pesticide spray: systemic movement is a crucial parameter for field application. To prove the dsRNA systemic movement in the phloem to the whole plant; locally defined spray was followed by *Sitobian* feeding on *T. aestivum* cvs stem, leaf, and root. RNAi biopesticides are gaining more importance as a substitute for chemicals with minimum side effects and a directed approach for a specific pest [45]. The systemic movement shows lac1 20, 40, and 60% expression in leaves, stem, and root (Fig 2C). shRNA moves to phloem apoplast; symplast, parenchyma, companion, and mesophyll cells are reported [45, 46]. However, the mechanism by which the sprayed RNA overcomes the apoplastic-symplastic barrier is yet to be discovered [46]. The importance of the lac gene in plant and insect defense is clear. The Salac had an important role in insect immunity through its exoskeleton hardening, and plant defense attenuation [10, 12–14]. The lac importance of insect resistance in cotton bollworms is also reported [11]. Salac1 silencing is linked to *S. avenae* polyphenol oxidase, and iron metabolism disruption with significant mortality [13]. The Salac1 identified in native grain aphid (ON703252) had a similarity to the copper oxidase binding domain. dsRNA assay resulted in 69% mortality 8D-post feeding; and 61% silencing of the Salac gene (Fig 2B). Peng et al. [47] reported the role of copper-binding oxidase in insect viability, living, and fertility. But still, there is little understanding of lac1 role in insect immunity-related processes [11]. The

research also focused on dsRNA regulation based on target identification and risk identification by ERNAi. Off-targeting can be a problem in defense responses. So siRNA targets were predicted using ERNAi *in-silico* approach with pea-aphid as a reference with no off-targets (S1 Table).

*Agrobacterium*-mediated transformation is employed for *in-vitro* and *in-planta* transformation. The transformation efficacy is poor in monocots and grains like wheat, maize, and rice: which are reluctant to *Agrobacterium* infection was improved by acetosyringone [19, 48]. *Triticum* cvs transformation was optimized with 250 µM acetosyringone, improving efficiency from 10–70% (Fig 6A). *In-planta* transformation, gave higher transformation efficiencies with wheat cvs Galaxy 2012, Anaj 2017, and Punjab (5–16%) (Fig 6C). While *in vitro* transformation efficiency was found quite low in the range of 0.2–1.5% (Fig 6B). Minimum aphid survival was observed in transgenic *Triticum* cvs Galaxy 2012, followed by Punjab, and Anaj 2017 (22–52%) compared to +ve control (Zincol 2016) (20%) (Fig 7B). 30–70% lac expression was observed in transgenic wheat cvs (Fig 7A). The difference in transgene expression was due to variations in genotypes. Immature *Arabidopsis thaliana* flowers are most frequently transformed via the *Agrobacterium*-mediated flower-dipping technique targeting ovule cells [49]. For improved insect resistance in chickpeas, cry1Ac was transformed into pRD400 by *A. tumefaciens in-vitro* transformation, with 3.6% transformation efficiency [50]. A codon-optimized chimeric Bt gene was transformed with 0.076% transformation efficiency through *in-vitro* shoot regeneration against insects [51]. *In planta*, and particle bombardment *Agrobacterium*-mediated transformation is reported in wheat [52]. Cry1Ac, Cry2A, and cp4EPSPS genes were transformed to improve insect resistance in cotton through shoot tip, a direct gene transfer method was 1.2 and 1.5% for Cry2A, and CP4EPSPS respectively [53]. Transgenic cotton against insects revealed 0.93% transformation efficiency by pistil dip transformation approach, an *in planta* transformation strategy [54]. Results show a higher *in planta* transformation efficiency compared to *in-vitro* transformation.

## Conclusion

Lac1 had been identified as an important RNAi target in *S. avenae*. Potential siRNA targets with no off target had been found by ERNAi. Phylogenetic analysis shows 90% homology to *A. pisum*, and *M. persicae*. Lac expression was reduced by 61% by dsRNA assay. The dsRNA lac exogenous application had shown its systemic movement; from leaf to stem and root. The lac expression reduction from leaf (20%) to stem (40%), and root (60%); 9-13D-post spray was calculated. Lac ihpRNA construct successfully knocks down the lac expression in aphids post-transgenic *T. aestivum* cvs feeding, with 48–78% mortality. *In planta* transformation efficiency of 5–16% was found in *Triticum* cvs. Galaxy 2012, Anaj 2017, and Punjab.

## Supporting information

**S1 File. The cloning system based on RNAi GG vector construct using type IIS restriction enzyme BsaI is deviced.** Vector containing a Pdk intron region; form a hairpin loop structure, two copies of the ccdB gene flanked by BsaI restriction sites. For SaLac1 gene insertion to the RNAi-GG vector, the ccdB genes are removed with BsaI; identifying sequences upstream and downstream of the pdk intron. Subsequently; adaptor sequences specific to the RNAi-GG vector are then added to Salac gene, enabling its insertion in both sense and antisense directions on either side of the Pdk intron region. The resulting hairpin loop incorporates Salac; triggers the SaLac silencing through RNA interference.
(DNA)

**S2 File. Laccase primer efficiency calculation by sample serial dilutions.** CT values were plotted against the log of sample quantity.
(XLSX)

**S1 Table. List of siRNA and dsRNA targets for Laccase gene by ERNAi (Boutros lab, E-RNAi-Version 3.2; Horn and Boutros 2010) (On page 7 before discussion).**
(DOCX)

**S1 Raw images.**
(PDF)

# Acknowledgments

The authors are thankful to Dr. Shaukat Ali for his correction in English Language.

# Author Contributions

**Conceptualization:** Amber Afroz, Nadia Zeeshan.

**Data curation:** Amber Afroz, Waheed Chatha.

**Formal analysis:** Amber Afroz, Muhammad Azmat Ullah Khan, Nazia Rehman.

**Funding acquisition:** Amber Afroz.

**Investigation:** Umer Rashid.

**Methodology:** Asma Rafique, Umer Rashid, Waheed Chatha.

**Software:** Muhammad Azmat Ullah Khan, Muhammad Irfan.

**Supervision:** Amber Afroz, Nadia Zeeshan.

**Validation:** Asma Rafique.

**Visualization:** Muhammad Irfan, Muhammad Ramzan Khan.

**Writing – original draft:** Asma Rafique.

**Writing – review & editing:** Amber Afroz, Muhammad Ramzan Khan, Nazia Rehman.

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
