## [Decision Letter · Decision Letter 0]

20 Mar 2023

PONE-D-23-00481Golden Gate assembly for Sitobion avenae laccase silencing and its application as a biopesticidePLOS ONE

Dear Dr. Afroz,

Thank you for submitting your manuscript to PLOS ONE. After careful consideration, we feel that it has merit but does not fully meet PLOS ONE’s publication criteria as it currently stands. Therefore, we invite you to submit a revised version of the manuscript that addresses the points raised during the review process.

The article "Golden Gate assembly for Sitobion avenae laccase silencing and its application as a biopesticide" has merit, and needs to address the points raised by the reviewers. Authors are suggested to include original gel images for the gels given in this article. The detailed review reports are attached for the authors information.

We look forward to receiving your revised manuscript.

Kind regards,

Basavantraya N. Devanna, PhD

Academic Editor

PLOS ONE

Journal Requirements:

4. We note that Figures 1 and 7 in your submission contain copyrighted images. All PLOS content is published under the Creative Commons Attribution License (CC BY 4.0), which means that the manuscript, images, and Supporting Information files will be freely available online, and any third party is permitted to access, download, copy, distribute, and use these materials in any way, even commercially, with proper attribution. For more information, see our copyright guidelines: http://journals.plos.org/plosone/s/licenses-and-copyright.

a. You may seek permission from the original copyright holder of Figures 1 and 7 to publish the content specifically under the CC BY 4.0 license. 

Additional Editor Comments:

This article has merit, however the language and contents needs to be improved for better readability.

Please find the detailed queries from the reviewers.

Reviewers' comments:

Reviewer's Responses to Questions

**Comments to the Author**

1. Is the manuscript technically sound, and do the data support the conclusions?

Reviewer #1: Yes

Reviewer #2: Yes

2. Has the statistical analysis been performed appropriately and rigorously? 

Reviewer #1: Yes

Reviewer #2: Yes

3. Have the authors made all data underlying the findings in their manuscript fully available?

Reviewer #1: Yes

Reviewer #2: Yes

4. Is the manuscript presented in an intelligible fashion and written in standard English?

Reviewer #1: Yes

Reviewer #2: Yes

5. Review Comments to the Author

Reviewer #1: The manuscript entitled "Golden Gate assembly for Sitobion avenae laccase silencing and its application as a

biopesticide" reports a details investigation of RNAi strategy to control Sitobion avenae. The manuscript can be accepted in its present form, however a extensive proofreading is suggested to improve the readability of the manuscript.

Reviewer #2: In this manuscript, author evaluated efficacy of Lac. The manuscript is well written and meet the quality of this journal. I would recommend this manuscript for publication with minor revision.

Comments:

Why only authors selected dsRNA 20 ng μL−1 con. for bio-efficacy study?

1. The current title will not fit and need to be change

2. What is the efficiency of RT-PCR and qRT-PCR primers? are these validated? If yes, data should be included.

3. Line number 187-204 need to be merged and re written

4. Line number 150-157 and 234-242 are seems to be same.

5. All sub title are inappropriate and follow first letter as upper and the rest lower case

6. Fig.4, 5 and 6D, are looks bad quality and edited. submit original gel images.

7. Author tested six different concentrations of Acetosyringone and how is the transformation efficiency calculated and data should submit either in table or graphical

6. PLOS authors have the option to publish the peer review history of their article (what does this mean?). If published, this will include your full peer review and any attached files.

Reviewer #1: No

Reviewer #2: **Yes: **Yugander Arra

---

## [Author Response · Author response to Decision Letter 0]

8 Apr 2023

Editorial Comments

Answer: The manuscript is according to PLOS ONE requirement; Crosschecked. 

2. In the Method section permits you obtained for the work. Please ensure you have included the full name of the authority that approved the field site access and, if no permits were required, a brief statement explaining why.

Answer: Work is performed in the Department of Biochemistry and Biotechnology, University of Gujrat, Gujrat Pakistan is added to the Methodology.

3. PLOS ONE now requires that authors provide the original uncropped and unadjusted images underlying all blot or gel results reported in a submission’s figures or Supporting Information files. This policy and the journal’s other requirements for blot/gel reporting and figure preparation are described in detail.

Answer: All original figures are attached (File Name: Original Figures).

Answer: Not Applicable. All figures are in original figures.

5. We note that Figures 1 and 7 in your submission contain copyrighted images. All PLOS content is published under the Creative Commons Attribution License (CC BY 4.0), which means that the manuscript, images, and Supporting Information files will be freely available online, and any third party is permitted to access, download, copy, distribute, and use these materials in any way, even commercially, with proper attribution.

Answer: Figures 1 and 7 were the original figures that seems to be similar to some previous report. The figures are replaced with new ones. 

6. Please include captions for your Supporting Information files at the end of your manuscript, and update any in-text citations to match accordingly.

Answer: Caption of supporting information is included & Highlighted.

Answer to the Reviewers 1

I am thankful to the reviewer for their valuable comments.

Comment 1. The manuscript must describe a technically sound piece of scientific research with data that supports the conclusions. Experiments must have been conducted rigorously, with appropriate controls, replication, and sample sizes. All data underlying the findings in their manuscript is fully available. The manuscript is presented in an intelligible fashion and written in standard English. 

The conclusions must be drawn appropriately based on the data presented. Statistical analysis has been performed appropriately and rigorously.

Answer: Thanks for the comments. The conclusion had been changed.

Reviewer #1: The manuscript entitled "Golden Gate assembly for Sitobion avenae laccase silencing and its application as a biopesticide" reports a details investigation of RNAi strategy to control Sitobion avenae. The manuscript can be accepted in its present form, however, extensive proofreading is suggested to improve the readability of the manuscript.

Answer: Proofreading is done and checked by Dr. Shaukat Ali. 

Answer to the Reviewers 2

Reviewer # 2: In this manuscript, the author evaluated the efficacy of Lac. The manuscript is well-written and meets the quality of this journal. I would recommend this manuscript for publication with minor revision.

Comments: 

1. Why did only authors select dsRNA 20 ng μL−1 con. for the bio-efficacy study?

Answer: dsRNA 20 ngL-1 had been reported in previous reports for dsRNA assay. It is previously checked in OBP8 in our previous reports. It is cited now (Line # 155, 177).

2. The current title will not fit and need to be changed.

Answer: The Title is changed from “Golden Gate assembly for Sitobion avenae laccase silencing and its application as a biopesticide” to

“Production of Sitobion avenae-resistant wheat using laccase as RNAi target and its systemic movement in Triticum aestivum post-dsRNA spray”

Short title: changed from “Laccase dsRNA for improved grain aphid resistance” to “Laccase silencing for improved grain aphid resistance”.

2. What is the efficiency of RT-PCR and qRT-PCR primers? are these validated? If yes, data should be included.

Answer: Primer validation data is included now in the paper (Supp File 2).

3. Line numbers 187-204 need to be merged and re-written.

Answer: Line Numbers 202-217 are merged and rewritten.

4. Line numbers: 150-157, and 234-242 seem to be the same.

Answer: 160-172 are retained and Line number 232-242 are reduced to two lines (246-247).

5. All subtitles are inappropriate and follow the first letter as upper and the rest as lowercase.

Answer: Changed throughout & highlighted. 

6. Fig.4, 5, and 6D, are looks bad quality and edited; submit original gel images.

Answer: Original Gel images are added.

7. Author tested six different concentrations of Acetosyringone and how the transformation efficiency calculated data should submit either in a table or graph.

Answer: The graph is included in the paper with acetosyringone affecting the transformation efficiency in Fig 6A. It is with the hygromycin selection the acetosyringone concentration is calculated. Transformation efficiency shown in the graph is based on selection.

---

## [Decision Letter · Decision Letter 1]

11 Apr 2023

Production of Sitobion avenae-resistant Triticum aestivum cvs using laccase as RNAi target and its systemic movement in wheat post dsRNA spray

PONE-D-23-00481R1

Dear Dr. Afroz,

We’re pleased to inform you that your manuscript has been judged scientifically suitable for publication and will be formally accepted for publication once it meets all outstanding technical requirements.

Kind regards,

Basavantraya N. Devanna, PhD

Academic Editor

PLOS ONE

Additional Editor Comments (optional):

The revised article "Production of Sitobion avenae-resistant Triticum aestivum cvs using laccase as RNAi target and its systemic movement in wheat post dsRNA spray" is much better than the original submission. Both quality and content of the MS is improved.

Reviewers' comments:

Reviewer's Responses to Questions

**Comments to the Author**

1. If the authors have adequately addressed your comments raised in a previous round of review and you feel that this manuscript is now acceptable for publication, you may indicate that here to bypass the “Comments to the Author” section, enter your conflict of interest statement in the “Confidential to Editor” section, and submit your "Accept" recommendation.

Reviewer #2: All comments have been addressed

2. Is the manuscript technically sound, and do the data support the conclusions?

Reviewer #2: Yes

3. Has the statistical analysis been performed appropriately and rigorously? 

Reviewer #2: Yes

4. Have the authors made all data underlying the findings in their manuscript fully available?

Reviewer #2: Yes

5. Is the manuscript presented in an intelligible fashion and written in standard English?

Reviewer #2: Yes

6. Review Comments to the Author

Reviewer #2: (No Response)

7. PLOS authors have the option to publish the peer review history of their article (what does this mean?). If published, this will include your full peer review and any attached files.

Reviewer #2: **Yes: **Yugander Arra

---

## [Editor Report · Acceptance letter]

2 May 2023

PONE-D-23-00481R1 

Production of *Sitobion avenae*-resistant *Triticum aestivum* cvs using laccase as RNAi target and its systemic movement in wheat post dsRNA spray 

Dear Dr. Afroz:

I'm pleased to inform you that your manuscript has been deemed suitable for publication in PLOS ONE. Congratulations! Your manuscript is now with our production department. 

Kind regards, 

on behalf of

Dr. Basavantraya N. Devanna 

Academic Editor

PLOS ONE